# Spatial distribution pattern and driving mechanism of tourist attractions in Gansu Province based on POI data

**Ruijuan Peng[1,2]ᵒ, Wanqianrong Gao[1]ᵒ***

1 Tourism College, Northwest Normal University, Lanzhou, Gansu, China, 2 Gansu Tourism Development Academy, Lanzhou, Gansu, China

ᵒ These authors contributed equally to this work.
* gaowanqianrong@163.com

## Abstract

The article utilizes POI (Point of Interest) data of tourist attractions in Gansu Province in 2021, adopts Moran's I and kernel density analysis to study the spatial distribution pattern of tourist attractions in Gansu Province, and uses spatial autoregressive modeling to explore the driving mechanism affecting their spatial distribution pattern. The results show that: (1) Gansu Province has a large number and rich types of tourist attractions, and there are differences in the number of different types of tourist attractions; (2) The spatial distribution pattern of different types of tourist attractions in different cities and towns shows the phenomenon of both agglomeration and dispersion, with a higher degree of agglomeration in the central and northwestern regions of the province and a lower degree of agglomeration in the southwestern and southeastern corners; (3) The overall spatial distribution pattern of tourist attractions shows the distribution characteristics of multi-core decentralized distribution, forming 8 core aggregation areas in the southeast of the province; (4) The article analyzes the driving mechanism of the spatial distribution pattern of tourist attractions in Gansu Province using the buffer zone and OLS models, and the results show that the natural environment, transportation location, national policies and socio-economics all have a positive impact on the distribution of tourist attractions.

**Data Availability Statement:** The vector map information of administrative boundaries of Gansu Province was obtained from the National Basic Geographic Information System database (http://www.ngcc.cn). The cross-sectional data of 14

## 1 Introduction

Tourist attractions are important carriers of tourism development and core elements of the tourism system, and their geographical location and spatial pattern play an important role in the rational development and high-quality development of tourism. Relevant studies on the spatial distribution pattern of tourism began in the 1960s, when scholars such as Christaller [1], Stang [2], Weaver [3] and Ali Movahed [4] used theories such as the location theory, the central place theory, the core-edge theory and the behavioral geography to study the spatial structure and pattern of the tourist sites. To explore the structural relationships and interactions that exist between the space of tourism activities, service systems and spatial patterns of

prefectures and cities were obtained from the Statistical Yearbook of Gansu Province and the Statistical Bulletin of National Economic and Social Development of 2021 of each prefecture and city.

**Funding:** Funding: (1) National Natural Science Fundation of China(42261034):Influence mechanism of multi-scale spatial structure on carbon emissions in the Western Valley city. (2) Higher Education Innovation Fund Project "Study on carbon emission and carbon carrying capacity measurement, spatial and temporal evolution and driving mechanism of tourism in the Yellow River Basin" (2023B-073).

**Competing interests:** The authors have declared that no competing interests exist.

tourism geography, as well as the spatial behavior patterns of tourists. In terms of research methodology. Scholars such as Hong Yang [5], Wen-Rong Pan [6], Degen Wang [7] and Zhang Shengrui [8] have explored and studied the temporal characteristics and spatial clustering characteristics of the spatial pattern of China's tourism economy, the spatial evolution and distribution of inbound tourism, the impact of accessibility to 338 HSR-connected cities and the spatial pattern of tourism development in the study area and the influencing factors using GIS technology. Scholars such as Prem Chhetri [9] and Tamara de la Mata [10] used spatial econometric techniques and gravity models to conduct an econometric analysis of the spatial pattern of tourism in the study area, the role of the urban economy on tourism employment and the trade flows between tourism-related sectors. Scholars such as Shien Zhong [11], Yeoman Ian [12] and Zhang [13] for the analysis of spatial patterns of regional tourism phenomena and tourism activities using global statistics and local spatial correlation indicators through the construction of new geographical frameworks, improved field models and network techniques.

As for data sources. Scholars such as P. C. Fore [14], Pearce Douglas G [15], Miguel Seguí-Llinás [16], and Bálint Kádár [17], by analyzing tourism brochures, civil aviation data at different scales and geo-referenced photographs from shared websites in the study area. An empirical study of the spatial pattern of tourism flows and tourism patterns, tourist attractions and spatial patterns, the evolution of the spatial pattern of demand for charter tourism and tourism systems. In terms of research area and object selection. Zhang Ai [18], Zhan Zirui [19] and Zhang Shengrui [20] selected national rural tourism key villages, Chinese national rural tourism towns and ethnic minority regions in northwest China to explore the spatial patterns and influencing factors of their rural tourism resources. Scholars such as Wang Yuewei [21], Chen Xuejun [22] and Zhao Junyuan [23] have studied the spatial pattern of tourism eco-efficiency in Inner Mongolia, the evolution of tourism flows in China's Chengdu-Chongqing Economic Circle and the spatial and temporal patterns and driving mechanisms of tourism eco-safety in the Yellow River Basin. For the study of the spatial pattern of tourism in Gansu Province, China. Scholars such as Wang Shuo [24], Ba Duoxun [25], Wang Yuli [26], and Zhao Honliang [27] used mathematical and statistical analysis methods and GIS spatial analysis tools to study the spatial structure of A-class tourist attractions, ethnic tourism resources, inbound tourism economy, and rural tourism agglomeration in Gansu Province and their differences.

The above studies have enriched the theory and method of tourism spatial pattern and formed a more complete theoretical foundation. However, in terms of the selection of research objects, scholars mostly choose some tourism resources in the study area as the object of analysis, lacking the research on the spatial pattern of regional tourism resources as a whole. In terms of theories and research methods, quantitative research methods such as "point-axis" system theory, GIS spatial analysis and mathematical statistics are mostly used to analyze the characteristics of spatial distribution patterns, but there is a lack of research methods that combine with geospatial and spatial big data, and a lack of research paths that combine quantitative research and qualitative analysis. From the current status of research on tourism resources in Gansu Province, scholars have only analyzed the spatial structure of some tourism resources such as A-class tourist attractions, ethnic tourism and rural tourism in Gansu Province, but less work has been carried out to study the spatial structure of tourist attractions in the province as a research object. Based on the POI (Point of Interest) data of the tourist attractions in Gansu Province in 2021, the paper sorts out and classifies the tourist attractions in the province according to the classification standard of tourism resources, combines the main resource types that each tourist attraction relies on, and follows the principle of objectivity, and applies Moran's I and kernel density analysis to visualize the spatial pattern of tourist attractions in Gansu Province and analyze its distribution pattern, and applies buffer zone analysis and OLS

spatial autoregressive model to dissect the driving factors that form its spatial pattern. The spatial pattern of tourist attractions in Gansu Province is visualized and analyzed, and the driving factors for the formation of their spatial pattern are analyzed using buffer analysis and OLS spatial autoregressive model. It provides a scientific basis for the future development and layout of tourist attractions in Gansu Province, which is of great significance in promoting the optimization and upgrading of tourism development in Gansu Province and helping the high-quality development of its tourism industry, and it can also provide a reference for the development of tourism in neighboring provinces and cities.

## 2 Regional overview

Gansu Province is located in the inland region of northwest China (Fig 1), with long and narrow topography, complex and diverse landscapes, stretching more than 1,600 kilometers from east to west, diverse climate and unique scenery. With a long history, Gansu Province is one of the major birthplaces of Chinese civilization and a key point of the ancient Silk Road. The province has 14 prefectures and cities, and is home to a multi-ethnic population.

## 3 Data sources and research methods

### 3.1 Data sources and processing

POI (Point of Interest) data refers to point data in electronic maps on the Internet, and is a technical term for geographic data points in GIS, which has the characteristics of high accuracy and fast update in real time compared with traditional statistics [28]. Tourist attractions are things that can attract tourists of any type and any form. Using POI data can effectively present the spatial location of tourist attractions and reflect the spatial structure of tourist attractions, which is conducive to improving the accuracy of spatial pattern research of tourist attractions. Amap is a domestic map navigation product with comprehensive location-based life services and rich information, and also has the "Triple A" qualification. It has deep POI characteristics. Therefore, this paper selects 14 cities and prefectures in Gansu Province as the crawling area on the open platform of Gaode Map, and uses "landscape", "tourism" and "attractions" as the search keywords to crawl a total of 4404 POI data, the crawled data contains name, category, latitude, longitude, address and other attributes, after screening, rejection and de-weighting finally retain 4092 valid data, with reference to the national standard of "Tourism Resources Classification, Survey and Evaluation (GB/T 18972–2017)", combined with the category attributes of each tourist attraction data crawled, the POI data of tourist attractions were classified into Architectural Facilities category, Geomorphological Landscape category, Biological Landscape category, Water Scenery category, Heritage Sites category and Human Activities category according to their resource characteristics (Table 1 and Fig 2). Administrative boundaries, rivers, and traffic vector data come from the 1:250,000 basic geographic data of China provided by the National Geomatics Center of China (https://www.ngcc.cn/ngcc/html/1/391/392/16114.html), the 250m Digital Terrain Elevation Model (DEM) data comes from Resource and Environmental Science Data Center of Chinese, Academy of Sciences (https://www.resdc.cn/data.aspx?DATAID=123), and these data are available for free. The cross-sectional data of 14 prefectures and cities were obtained from the Statistical Yearbook of Gansu Province and the Statistical Bulletin of National Economic and Social Development of 2021 of each prefecture and city. In addition, we carefully read the terms of service of the relevant platform to ensure that our use of data is fully in accordance with the agreement.

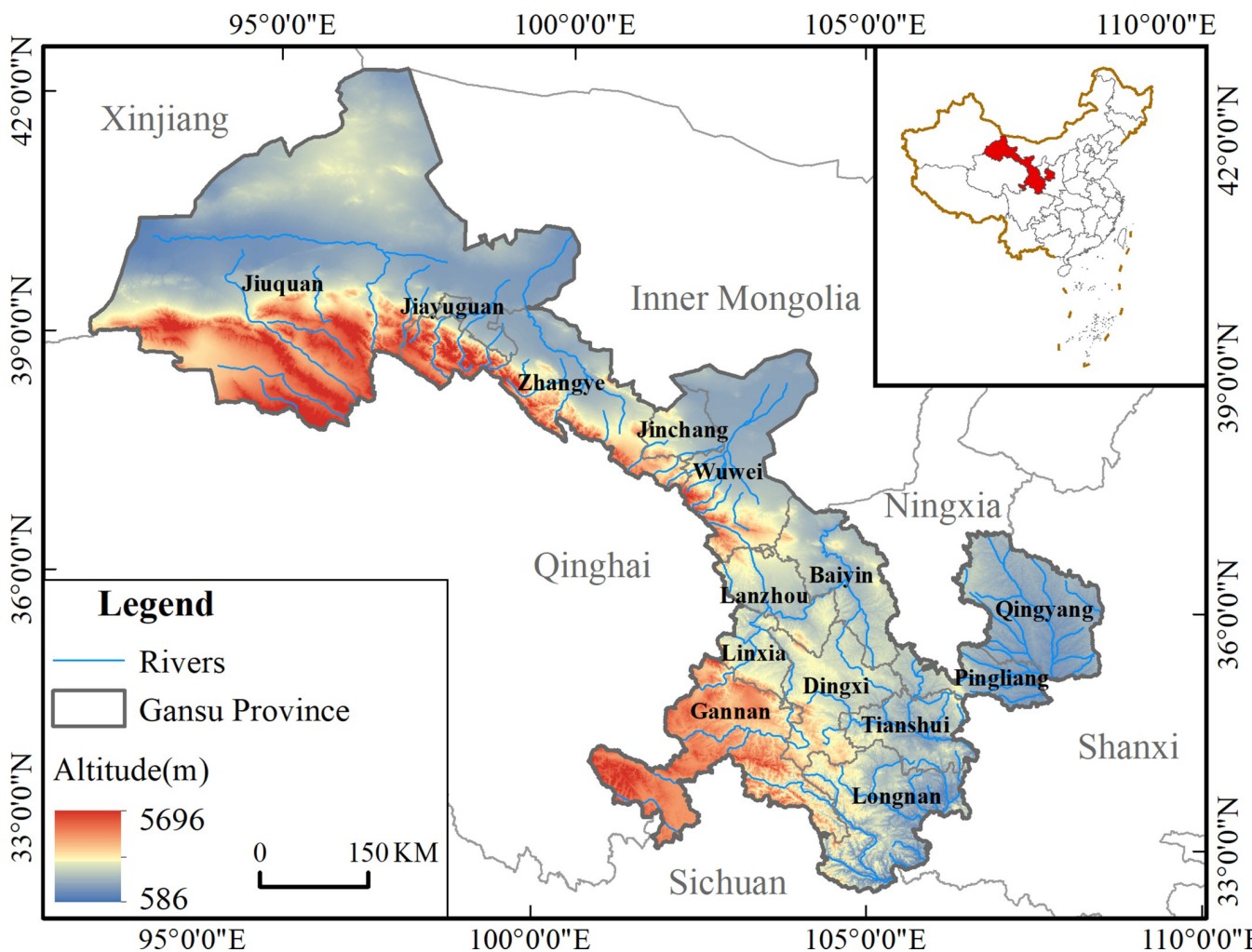

**Fig 1. Geographical location of Gansu Province and distribution map of various cities and cities.** Contains data from Contains information from National Geomatics Center of China (https://www.ngcc.cn/ngcc/html/1/391/392/16114.html) and Resource and Environmental Science Data Center of Chinese (https://www.resdc.cn/data.aspx?DATAID=123), they are freely available.

Table 1.  POI classification of Gansu tourist attractions.

| Types of tourist attractions | Specific POI data content | Quantity/pc | Proportion/% |
|---|---|---|---|
| Architectural facilities category | Comprehensive human tourism sites, single event venues, landscape buildings and accessory buildings, residential sites and communities, burial sites, transportation buildings, and waterworks buildings. | 2761 | 67.50 |
| Geomorphological landscape category | Comprehensive natural tourism sites, geological and geomorphological processes. | 427 | 10.40 |
| Biological landscape category | Trees, grasslands and meadows | 412 | 10.07 |
| Water scenery category | River sections, natural lakes and ponds, waterfalls, springs, snow and ice fields | 229 | 5.60 |
| Heritage sites category | Prehistoric human activity sites, socio-economic and cultural activity sites remains | 193 | 4.72 |
| Human activities category | Art, folk customs, modern festivals | 70 | 1.71 |

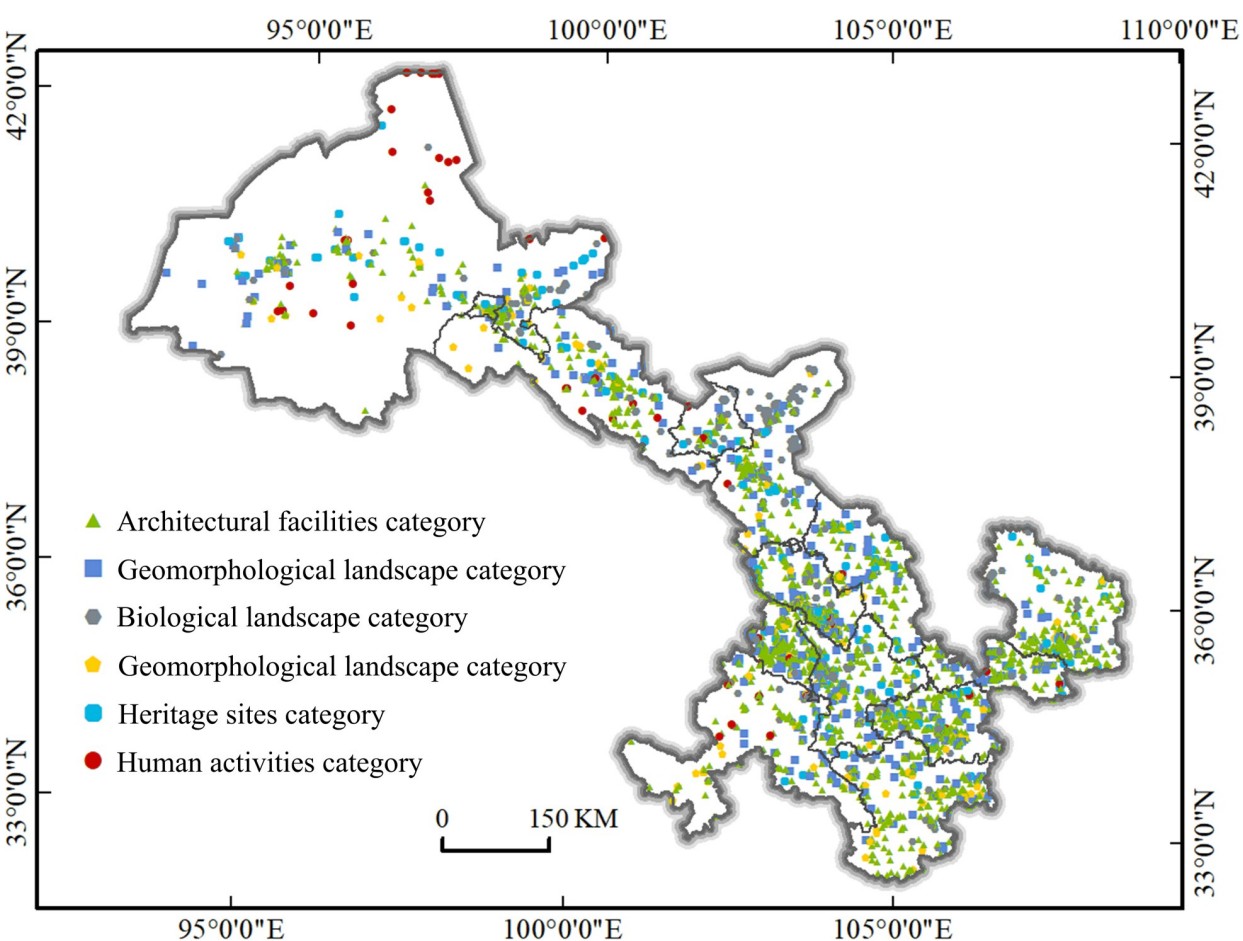

**Fig 2. Distribution map of tourist attractions in Gansu Province.** Contains data from Contains information from National Geomatics Center of China (https://www.ngcc.cn/ngcc/html/1/391/392/16114.html), they are freely available.

### 3.2 Research methodology

**3.2.1 Spatial autocorrelation analysis.** Spatial autocorrelation consists of global and local spatial autocorrelation. Global spatial autocorrelation uses Moran's I index to determine the spatial distribution characteristics of tourist attractions in the whole region; local spatial autocorrelation (Local Moran's I) visualizes the spatial distribution of local differences by drawing LISA agglomerative maps to further study the spatial heterogeneity of various tourist attractions in different regional [29].The local spatial autocorrelation (Local Moran's I) visualizes the spatial distribution of local differences by drawing LISA agglomerative maps to further study the spatial heterogeneity of various tourist attractions in different regions [30, 31].

$$I = \frac{n\sum_{i=1}^{n}\sum_{j=1}^{n}W_{ij}(X_i - \bar{X})(X_j - \bar{X})}{\sum_{i=1}^{n}\sum_{j=1}^{n}W_{ij}\sum_{i=1}^{n}(X_i - \bar{X})^2} = \frac{\sum_{i=1}^{n}\sum_{j\neq 1}^{n}W_{ij}(X_i - \bar{X})(X_j - \bar{X})}{S^2\sum_{i=1}^{n}\sum_{j\neq 1}^{n}W_{ij}} \tag{1}$$

$$I_i = \frac{(X_i - \bar{X})}{S^2}\sum_i W_{ij}(X_j - \bar{X}) \tag{2}$$

In Eq (1), is Moran's I, $n$ is the sample size, that is, the number of tourist attractions, when Moran's I $> 0$, it means that the class of tourist attractions presents significant spatial positive correlation, when Moran's I $= 0$, it means that the class of tourist attractions does not have spatial correlation, when Moran's I $< 0$, it means that the class of tourist attractions presents significant spatial negative correlation, Eq (2) is the formula of local spatial autocorrelation coefficient.

**3.2.2 Kernel density estimation.** Kernel density estimation (KDE) is an algorithm to calculate the spatial distribution of point data, which is used to calculate the density of elements in their surrounding neighborhoods and can reflect the spatial clustering of analysis targets. In this paper, we use the kernel density analysis method to analyze the spatial distribution of the overall and various types of tourist attractions in the whole range of Gansu Province, and the formula is as follows [32, 33].

$$f_n(x) = \frac{1}{nh}\sum_{i=1}^{n} k\left(\frac{x - x_i}{h}\right) \tag{3}$$

In the above equation, $k$ is the kernel function, $x$ is the location of the tourist attractions in Gansu Province, $x_i$ is the specific location in space of the tourist attractions in Gansu Province formed by $x$ as the center of the circle, $h$ is the radius, and $n$ indicates the sample size, that is, the number of tourist attractions.

**3.2.3 Spatial autoregressive model.** This paper utilizes the OLS (Ordinary least squares) regression model, also known as least squares [34, 35]. The POI data of tourist attractions in Gansu Province were imported into ArcGIS and spatially connected with the panel data characterizing the drivers so that they have unique IDs in each city and state. Spatial analysis was carried out with the help of Ordinary Least Squares (OLS tool) in the toolbox of spatial analysis, with the number of tourist attractions as the dependent variable, national policy, total tourism income, and disposable income of residents as the independent variables, with the following formulas:

$$y = \beta x + \mu \tag{4}$$

In the above equation, $y$ is the dependent variable, $x$ is the independent variable, $x$ is the regression coefficient of $\beta$, $\mu$ indicates the random error, and must obey the normal distribution.

# 4 Characteristics of spatial distribution pattern of tourist attractions in Gansu Province

## 4.1 Characterization of the level and type structure of tourist attractions

Grade A tourist attractions are high-quality representatives of tourism resources and an important hand in the high-quality development of the tourism industry [36]. According to the assessment standards of China's Ministry of Culture and Tourism for national Class A tourist attractions, as of December 31, 2022, there were 442 Class A tourist attractions in Gansu Province, including 7 Class 5A tourist attractions, 133 Class 4A tourist attractions, 232 Class 3A tourist attractions, 69 Class 2A tourist attractions, and 1 Class 1A tourist attraction (Table 2). In various cities and towns, there is an unbalanced distribution, with Jiuquan City having the largest number of 52 and Jiayuguan City having the smallest number of only seven. And the number of tourist attractions is mostly concentrated in 4A and 3A level, and the overall level shows the olive-shaped structure of "big in the middle and small at both ends".

**Table 2. Number of A-class tourist attractions in various cities of Gansu Province.**

| Municipalities | Rank | | | | | |
|---|---|---|---|---|---|---|
| | AAAAA | AAAA | AAA | AA | A | total |
| Jiuquan City | 1 | 15 | 32 | 3 | 1 | 52 |
| Tianshui City | 1 | 8 | 21 | 18 | / | 48 |
| Zhangye City | 1 | 22 | 23 | 2 | / | 48 |
| Longnan City | 1 | 17 | 21 | 5 | / | 44 |
| Lanzhou City | / | 8 | 25 | 7 | / | 40 |
| Linxia Prefecture | 1 | 9 | 24 | 3 | / | 37 |
| Pingliang City | 1 | 6 | 27 | 2 | / | 36 |
| Gannan Prefecture | / | 12 | 14 | 9 | / | 35 |
| Dingxi City | / | 10 | 7 | 12 | / | 29 |
| Wuwei City | / | 10 | 10 | 2 | / | 22 |
| Qingyang City | / | 5 | 13 | 3 | / | 21 |
| Baiyin City | / | 5 | 10 | / | / | 15 |
| Jinchang City | / | 3 | 2 | 3 | / | 8 |
| Jiayuguan City | 1 | 3 | 3 | / | / | 7 |
| Gansu Province | 7 | 133 | 232 | 69 | 1 | 442 |

A total of 4092 POI data of tourist attractions in Gansu Province were obtained through data cleaning, and there are 6 types of tourist attractions after classification, as shown in Fig 3. The province has the largest number of architectural facilities, with 2,761 attractions, accounting for 67.5% of the total, of which Lanzhou has the largest number, with 478 (17.3%), followed by Tianshui and Linxia, accounting for 13.4% and 11% respectively; the number of geo-landscape and bio-landscape tourist attractions are in second and third place, with 427 (10.4%) and 412 (10.07%), of which the largest number of geomantic landscape tourist attractions is in Lanzhou City, with 62, and the largest number of biological landscape tourist attractions is in

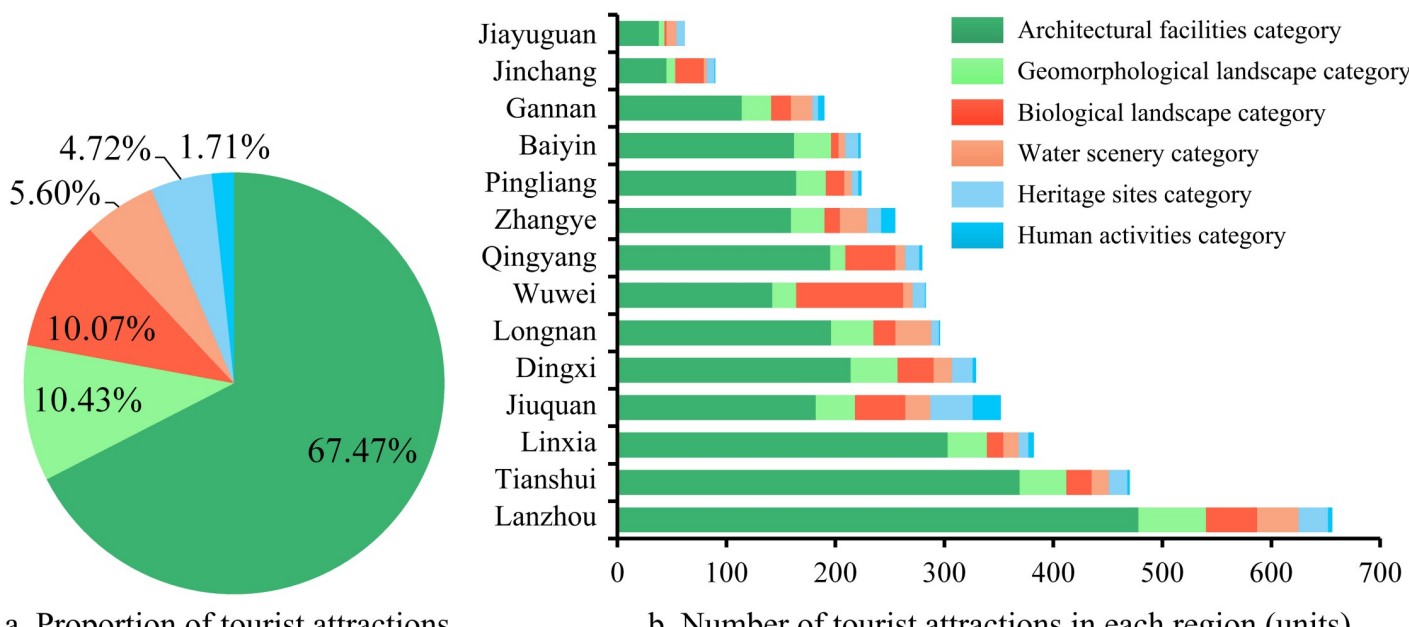

a. Proportion of tourist attractions

b. Number of tourist attractions in each region (units)

**Fig 3. The number and proportion of types of tourist attractions in cities and prefectures in Gansu Province.**

Wuwei City, with 98; the number of water scenery and ruins tourist attractions are in the fourth and fifth place, with 229 (5.6%) and 193 (4.72%), respectively, the largest number of water scenery tourist attractions in Lanzhou City, with 38, the most number of ruins and relics in Jiuquan City, 39; humanistic activities in the province's least number of tourist attractions in the type of tourist attractions, only 70, accounting for only 1.71% of the number of tourist attractions in the province, and Gansu Province's colorful cultural resources characteristics show the phenomenon of misalignment.

## 4.2 Characteristics of spatial distribution of tourist attractions

**4.2.1 Overall distribution pattern of tourist attractions.** *Global spatial autocorrelation of tourist attractions*. The number of different types of tourist attractions located within each city and state was extracted separately and the Moran's I index was measured in turn, and the results of the calculation are shown in Table 3. The overall tourist attractions in Gansu Province show spatial clustering characteristics, in addition, the spatial pattern of the remaining five types of tourist attractions show obvious clustering attributes, except for the insignificant clustering trend of the ruins and relics type of tourist attractions. Among them, the Moran's I index of human activities tourist attractions is the highest, reaching 0.235, indicating that the spatial clustering characteristics of this type of tourist attractions are the most significant; secondly, the Moran's I index of biological landscape and architectural facilities tourist attractions is also relatively high among the six types of tourist attractions, while the Moran's I index of water scenery and geomorphological landscape Only the Moran's I index of the heritage tourist attractions did not show obvious clustering phenomenon.

*Local spatial autocorrelation in tourist attractions*. ArcGIS software was used to further measure the local spatial autocorrelation of the corresponding number of tourist attractions extracted from the classification to each district (city) and county administrative area, and to draw the LISA clustering map of tourist attractions with obvious clustering characteristics of Moran's I index (Fig 4), and to classify the spatial association types of tourist attractions into high-high, high-low, low-high and low-low clustering areas, so as to analyze the spatial association characteristics of each type of tourist attractions. As can be seen from Fig 4(A), the high-high agglomeration area of tourist attractions in Gansu Province falls in Chengguan District, Qilihe District, Yuzhong County and Gangu County, Tianshui City, indicating that the above-mentioned areas have high agglomeration of their own tourist attractions and high agglomeration of tourist attractions in the surrounding areas; the low-high agglomeration area falls in Gaolan County, Lanzhou City, indicating that the agglomeration of tourist attractions in the region itself is low, but the agglomeration of tourist attractions in the surrounding areas is high, which is influenced by the "siphon effect" of the surrounding high-high agglomeration areas; the low-low agglomeration area is Luqu County, Xiahe City and Zhuoni County, Gannan Prefecture, which is influenced by the density of tourist attractions in the region itself and the surrounding areas. -The low-low agglomeration area is in Luqu County, Xiahe City and Zhuoni County, Gannan Prefecture, which are agglomerations with low density values of tourist attractions in the region itself and in the surrounding areas; in addition, the high-high agglomeration areas of tourist attractions of architectural facilities and geomantic landscape are in Lanzhou City, Tianshui City and its In addition, the high-high agglomerations of architectural facilities and geomorphic landscape tourist attractions fall in Lanzhou City, Tianshui City and their surrounding areas, the high-low agglomerations are in Dunhuang City, the low-high agglomerations fall in Longnan City and Baiyin City and other places, the low-low agglomerations fall in Gannan Prefecture and Qingyang City and other places; the high-high agglomerations of biological landscape, water scenery and humanistic activities tourist

**Table 3. Discrete characteristics of aggregation of various types of tourist attractions in Gansu Province.**

| Types of tourist attractions | Spatial aggregation of discrete features | | |
|---|---|---|---|
| | Global Moran's I | P-value | Z-value |
| Architectural facilities category | 0.183726 | 0.003071 | 2.9660511 |
| Geomorphological landscape category | 0.044966 | 0.068858 | 1.819353 |
| Biological landscape category | 0.160221 | 0.002549 | 3.017469 |
| Water scenery category | 0.158644 | 0.011222 | 2.53571 |
| Heritage sites category | 0.057068 | 0.284324 | 1.070656 |
| Human activities category | 0.23521 | 0 | 4.998084 |
| All Tourist Attractions | 0.176028 | 0.004664 | 2.829381 |

attractions fall in the five cities of Hexi, the high-low agglomerations are in Longnan City, Tianshui City and Qingyang City and other places, the low-high agglomerations fall in Zhangye City, Jiuquan City and parts of Lanzhou City areas, and low-low agglomeration areas are in Linxia, Gannan, Pingliang and Qingyang; the above results reflect that there are different differences in the agglomeration areas of various types of tourist attractions, among which Lanzhou and Tianshui have the most obvious phenomenon of agglomeration characteristics of tourist attractions. In addition, the central and northwestern areas of the province have a higher degree of agglomeration, while the southwestern and southeastern corners have a lower

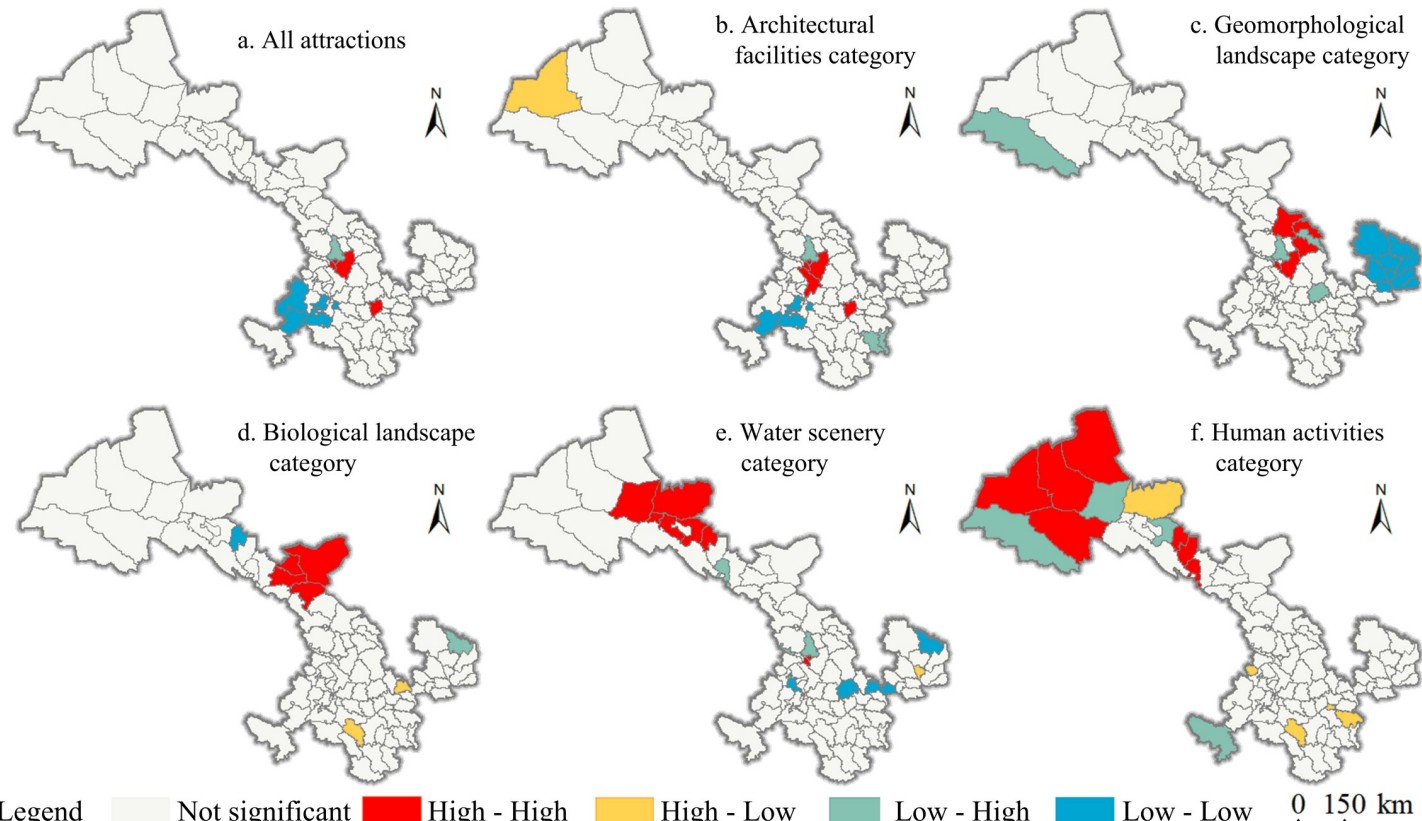

**Fig 4. LISA agglomeration figure of various types of tourist attractions in Gansu Province.** Contains data from Contains information from National Geomatics Center of China (https://www.ngcc.cn/ngcc/html/1/391/392/16114.html), they are freely available.

degree of agglomeration, showing the overall distribution characteristics of both agglomeration and dispersion.

**4.2.2 Differences in the distribution of the density of tourist attractions.** The above study shows that there is an obvious agglomeration of tourist attractions in Gansu Province within the administrative regions of each district (city) and county, and the results of nuclear density analysis of tourist attractions with the provincial domain as the study area are shown in Fig 5, where the spatial agglomeration of tourist attractions in different regions differs, and the overall spatial distribution pattern shows the distribution characteristics of multi-core dispersion. In the southeastern region, there are eight agglomerations in Lanzhou, Tianshui, Linxia, Dunhuang, Zhangye, Jiuquan-Jiayuguan, Wuwei and Qingyang, among which Lanzhou, Tianshui and Linxia have the highest density and form the core agglomerations with the highest degree of agglomeration; in Dunhuang, Zhangye, Jiuquan-Jiayuguan, Wuwei and Qingyang, there are secondary agglomerations with relatively low degree of agglomeration; in the northwestern region, there are scattered characteristics.

The common feature of the spatial distribution of various types of tourist attractions is that there are high-density areas in Lanzhou City, forming a "core-edge" spatial distribution feature. The high-density areas of architectural facilities and geological landscape tourist attractions are widely distributed in the province and mainly concentrated in the Hexi Corridor and the northeastern part of the province. The high-density areas of architectural facilities tourist attractions are mainly located in the political, economic and cultural centers of Lanzhou City, Tianshui City, Linxia Prefecture, Zhangye City, Wuwei City, Pingliang City, Qingyang City and Jiuquan City; the high-density areas of geological landscape tourist attractions are mainly located in Lanzhou City, Linxia Prefecture, Zhangye City, Baiyin City and Jiuquan City, which are rich in geomorphological types; Biological landscape tourist attractions in Wuwei City, Jinchang City, Jiuquan City, Lanzhou City, Dingxi City and part of Qingyang City to form a high-density area, most of the above areas are located in the edge of the desert, in order to effectively curb the sandy land, the local government planted trees, forming a number of unique biological landscape; water scenery and ruins tourist attractions clustered in a similar degree, mainly in Gansu Province, the western corridor and the northwest region. The high-density areas of water scenery tourist attractions are mainly concentrated in Lanzhou City, Linxia Prefecture, Zhangye City, Tianshui City, Longnan City, Jiuquan City and Jiayuguan City and other areas rich in water resources, and ruins and relics tourist attractions are densely distributed in Lanzhou City and the five cities of Hexi, the above-mentioned areas are the node cities of the ancient Silk Road and must pass through, with deep historical and cultural deposits; The spatial pattern of tourist attractions in the water scenery category and the ruins category have similar degrees of concentration and are mainly distributed in the Hexi Corridor and the northwest region of Gansu Province; the spatial pattern of tourist attractions in the humanistic activities category has obvious differences in the degree of concentration, with high-density areas in Jiuquan and Zhangye, concentrated in the northwestern minority areas or autonomous counties with rich and diversified cultural characteristics.

## 4.3 Analysis of the mechanism of the spatial distribution pattern of tourist attractions

**4.3.1 National policy level.** The construction and development of tourist attractions can not be separated from the policy support of the local government, regional tourism policy is an important driving force to build a good spatial pattern of tourist attractions [37]. Some areas in Gansu Province in the early stages of tourism development lack of certain macro-control, such as some ethnic areas with low levels of economic development, poor transportation

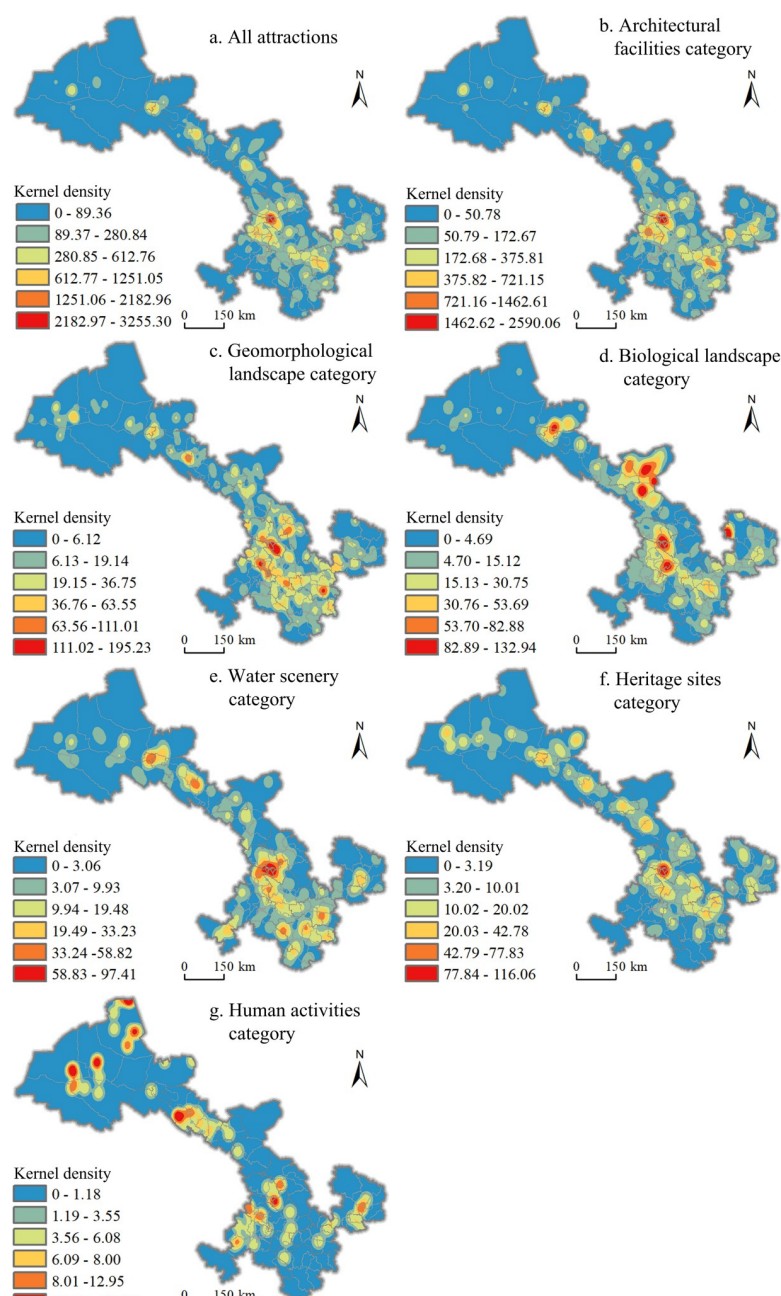

**Fig 5. Kernel density of various tourist attractions in Gansu Province.** Contains data from Contains information from National Geomatics Center of China (https://www.ngcc.cn/ngcc/html/1/391/392/16114.html), they are freely available.

location conditions, but very rich tourism resources, the construction of tourist attractions uneven development. The gradual improvement of tourism-related poverty alleviation policies has led to the development of tourism in poor areas with rich tourism resources. Gannan Prefecture, Linxia Prefecture and Jiuquan City Minority Autonomous County have introduced local policies to implement the development strategy of "tourism development", and through the adjustment of industrial structure, tourism has been cultivated into a pillar industry and a

new economic growth point for the whole state (county). The above measures have largely increased the number of regional tourist attractions and enriched the spatial pattern of tourist attractions in Gansu Province. In recent years, in order to implement the rural revitalization strategy, Gansu Province has given full play to the poverty alleviation advantages of tourism, relying on the rich farming cultural resources, and created 8 rural tourism demonstration counties and 60 model villages of cultural tourism revitalization villages in 2021 to promote the high-quality development of rural tourism, and the proportion of rural tourism resources in the tourist attractions in Gansu Province has gradually increased (Fig 6).

**4.3.2 Socio-economic dimensions.** The overall distribution of tourist attractions in Gansu Province shows the coexistence of agglomeration and dispersion, and the differences in the density distribution of various types of tourist attractions show that the most concentrated areas of tourist attractions are located in areas with high levels of economic development, such as Lanzhou City and Tianshui City, indicating that the regional economic level has an important influence on the development of tourist attractions. A good regional economic environment is the basis for the development of regional tourism, and a region with a high level of regional economic development can provide financial security for the development and construction of tourist attraction infrastructure [38]. And vice versa, tourist attractions as the core foundation of tourism development, its spatial pattern can feed the regional tourism industry and regional economic development, increase the regional gross domestic product and per capita income, etc., improve the consumption level of the residents in the area, to a certain extent, to provide sufficient source market for tourist attractions. In addition, regions with higher levels of economic development are more attractive to tourists and are an important driver of consumption for tourist attractions.

**4.3.3 Natural environment dimension.** Gansu Province is located at the intersection of three natural regions in China and has the geographical characteristics of all three natural regions, with huge differences in the geographical environment of different regions, and is also located at the intersection of the Qinghai-Tibet Plateau, Loess Plateau and Inner Mongolia Plateau, with a very complex topography. Gansu Province has four climate types, the most in

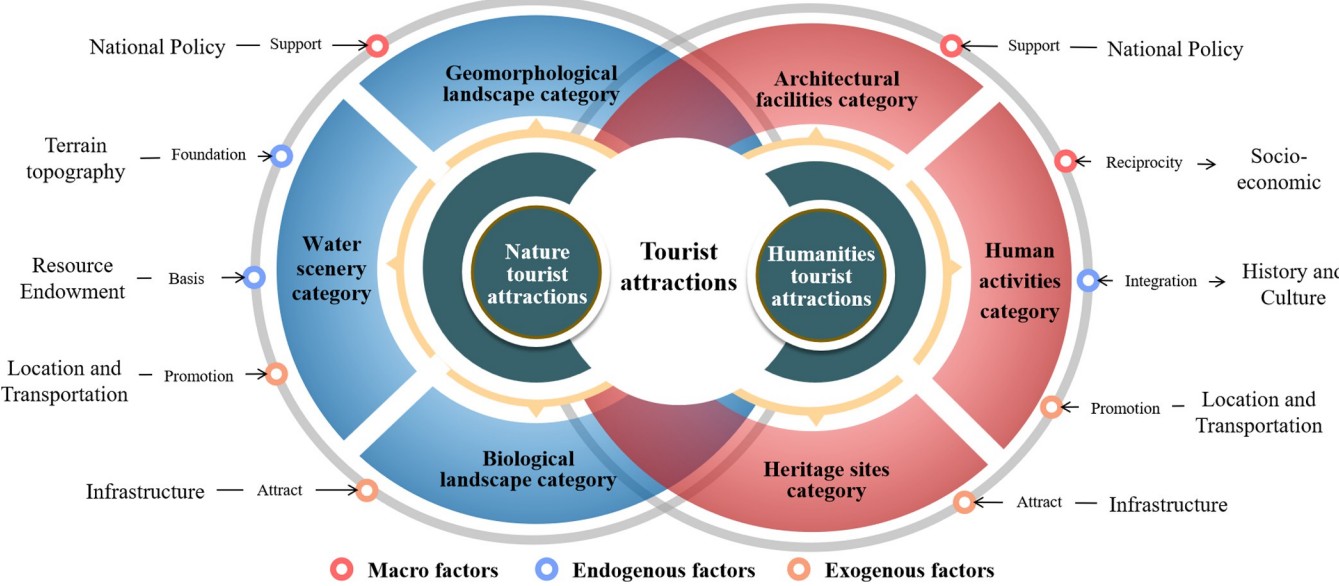

**Fig 6. Mechanism of influencing factors on spatial distribution pattern of tourist attractions in Gansu Province.**

China, and contains all of China's wet and dry zones; the province's rivers are divided into three major basins: the Yellow River, the Yangtze River and the Inland River, which contain nine water systems. Most natural resource-based tourist attractions are planned and developed based on the natural geography of the region itself, with topography and river hydrology being the main conditions for the formation of geomorphic landscapes and water scenery type tourist attractions [39]. For example, Lanzhou City and Tianshui City topographic features are narrow and long, showing "two mountains sandwiched by a river", "two mountains sandwiched by a ditch" spatial pattern, in the geomorphology has a natural advantage, the formation of a relatively rich geomorphological landscape of tourist attractions. Lanzhou City is the only one yellow river through the city of the provincial capital city, its territory by the main stream of the yellow river and its tributaries, Longnan City has Jialing River, Bailong River, Baishui River, West Hanshui four water system, the number of rivers, the annual runoff is large, the above two cities rely on the obvious advantages of river hydrology, the development and construction of a certain number of water scenery class tourist attractions. The complex and diverse climate types have created a diversity of biological resources in Gansu Province, forming a relatively rich biological landscape type of tourist attractions. Wuwei City straddles the Qilian Mountains, and the Qilian Mountain Nature Reserve in its territory is a national forest ecosystem type natural environment reserve and the largest forest ecosystem and wildlife type reserve in Gansu Province, Minqin County and Gulang County are located at the southern end of the Tengri Desert, and Wuwei City has planted a large number of plantation forests for desertification control, in addition to the Binggou River in Qilian Town, Tianzhu Tibetan Autonomous County, Wuwei City The scenic area contains natural scenery such as snow-capped mountains, heavenly pools, waterfalls, forests, grasslands and rivers. All of the above can reflect that the natural environment largely determines the number and spatial pattern of some types of tourist attractions.

**4.3.4 Cultural contextual dimension.** Regional culture is the core element of the integrated development of cultural tourism, and is crucial to the formation and development of tourist attractions in the category of heritage sites and humanistic activities [40]. Gansu Province is a large cultural province, in addition to the rich farming culture, there are Silk Road culture, Dunhuang culture, ancestral lineage culture, Yellow River culture, Great Wall culture, red culture, cave culture and space culture and other cultural brands, Gansu Province around its own unique cultural resources, and actively promote the integration of cultural tourism development, by 2025 will complete the construction of the Chinese civilization heritage innovation zone planning tasks, the Great Wall, the Long March, the Yellow River three National cultural parks are basically completed. But at this stage the number of humanities activities can be seen in the type of tourist attractions, Gansu Province, the development and utilization of cultural resources is still relatively scarce, failing to transform the cultural advantages into a strong cultural tourism industry.

**4.3.5 Location and transportation dimensions.** From the differences in the distribution of the density of various types of tourist attractions, it can be seen that the most concentrated tourist attractions in Gansu Province, Lanzhou City, Tianshui City and Dingxi City, etc. are located in the node of the main transportation arteries, location conditions are superior. Other areas with a lower level of transportation construction, such as Gannan Prefecture and Jinchang City, have not formed a more complete road transportation system, and the distribution of tourist attractions is more dispersed, without forming a cluster area. The above analysis can fully show that the development of regional transportation has an important impact on the formation of spatial patterns of tourist attractions, transportation is an important link between the source and destination [41], Gansu Province, the complex geomorphological environment, the difficulty of regional transportation access to a large extent determines the formation and

development of tourist attractions. The greater the accessibility of tourist attractions in regions with higher levels of accessibility, the greater the ease of access for tourists to visit, and the level of accessibility affects the choice of destination and mode of travel for tourists. Good transportation location conditions have become an advantage for the construction and development of regional tourist attractions, so more tourism developers will choose to carry out the development and construction of tourist attractions in areas with better location conditions, gradually leading to the alienation of the spatial pattern of tourist attractions in the region.

**4.3.6 Tourism services dimension.** With the booming development of tourist attractions, the quality of tourism services has become a powerful grip to enhance the competitiveness of tourist attractions [42]. Therefore, the level of tourism service facilities and service personnel affects the long-term development of tourist attractions and has a certain impact on the formation of the spatial distribution pattern of tourist attractions. According to Maslow's Hierarchy of Needs theory, after satisfying physiological and security needs, tourists need to obtain a sense of belonging and respect. High-quality tourism accommodation facilities can bring tourists a sense of belonging to home, while good services can complement the psychological care for tourists in tourism activities. A high standard of tourism services can attract potential tourists and also increase the rate of repeat visits by tourists, which further influences the high quality development of tourist attractions.

# 5 Study on the driving mechanism of the spatial distribution pattern of tourist attractions in Gansu Province

## 5.1 Identification of drivers

From the above analysis, it can be seen that the spatial distribution pattern of tourist attractions is affected by a variety of driving factors, this paper refers to the existing research [8, 43], combined with the current situation of the development of tourism in Gansu Province, taking into account the consistency of the statistical caliber of the panel data and accessibility, and analyzes the influencing mechanism of the spatial distribution pattern of tourist attractions in Gansu Province mainly from the perspective of the natural factors, transportation location and socio-economic three aspects.

## 5.2 Tourist attractions and the natural environment

Natural geography is an important foundation for the formation of tourist attractions, and this paper selects two natural environment factors, elevation and hydrology, for analysis. The spatial distribution map of tourist attractions in Gansu Province is superimposed with the topographic map, and the buffer zone analysis is carried out on the rivers above grade 3 in Gansu Province with an interval of 5km (Fig 7).

Fig 7 shows that, in terms of topographic elevation. Gansu Province has complex geomorphologic types with great differences in elevation, with heights gradually decreasing from southwest to northeast. Tourist attractions distribution can be divided into four altitude gradient, the western border of the Qilian Mountains as high as 4,000 to 6,000 meters above sea level, the distribution of tourist attractions sparse. Gannan Plateau region at an average elevation of about 3,000 meters above sea level, the distribution of tourist attractions are also fewer; Northwest Corridor at a relatively low elevation, tourist attractions from the northwest to the southeast direction of the gradual distribution of dense; the central Loess Plateau and the south of the Longnan mountainous terrain with the right elevation, mild climate, the distribution of tourist attractions showed obvious characteristics of the agglomeration. In terms of river hydrology. Gansu Province is rich in water resources, with the main river systems divided

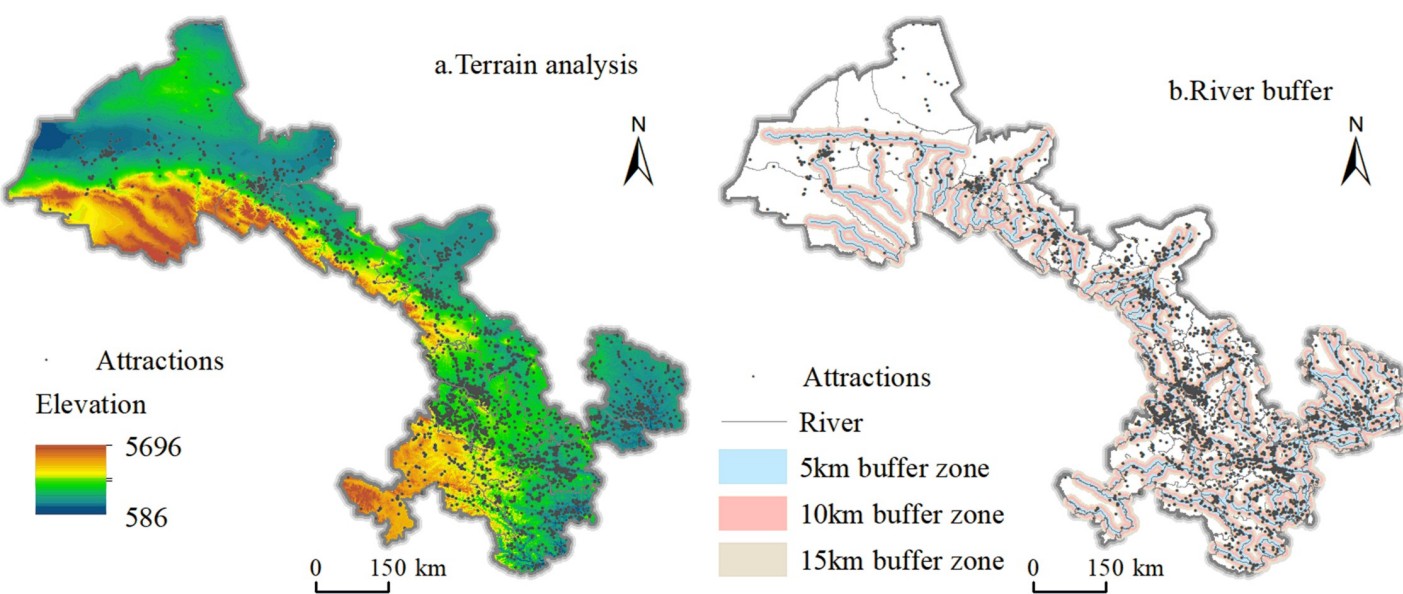

**Fig 7. Distribution of tourist attractions in Gansu Province coupled with topography and river buffers.** Contains data from Contains information from National Geomatics Center of China (https://www.ngcc.cn/ngcc/html/1/391/392/16114.html) and Resource and Environmental Science Data Center of Chinese (https://www.resdc.cn/data.aspx?DATAID=123), they are freely available.

into the Inland River, the Yellow River and the Yangtze River. In the northwestern part of the province, the climate is dry and the number of rivers is scarce, and the glacial meltwater of the Qilian Mountains is the main recharge source of the rivers, so the tourist attractions are distributed along the rivers. The Yellow River tributaries, Weihe River and Jinghe River, as well as the Jialing River, flow through the area where the distribution of the tourist attractions is

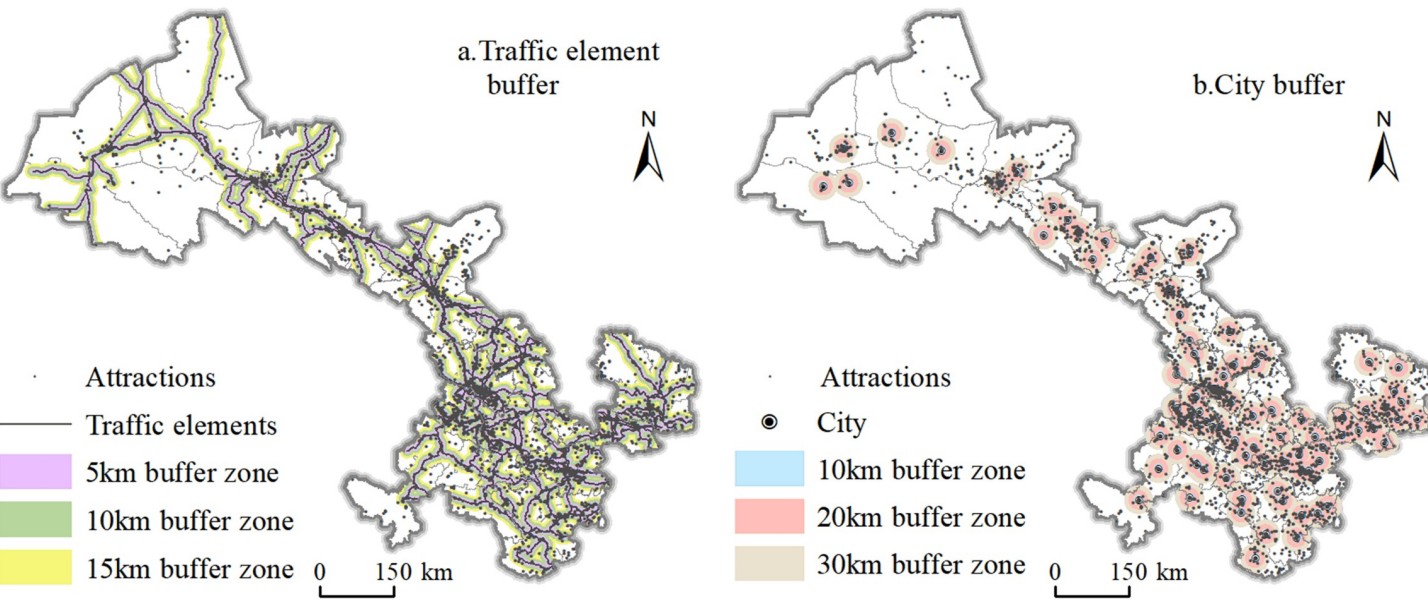

**Fig 8. Distribution of tourist attractions in Gansu Province traffic and urban buffer zone.** Contains data from Contains information from National Geomatics Center of China (https://www.ngcc.cn/ngcc/html/1/391/392/16114.html) and Resource and Environmental Science Data Center of Chinese (https://www.resdc.cn/data.aspx?DATAID=123)), they are freely available.

significantly clustered, and there are 2,159 tourist attractions in the 5-km buffer zone, which accounts for 52.8% of the total number of tourist attractions. The above analysis shows that natural factors have a significant influence on the distribution of tourist attractions.

## 5.3 Tourist attractions and transportation location

Transportation is an important channel connecting tourists and tourist destinations, and the construction and development of tourist attractions are also affected by the advantages and disadvantages of location. The railroads, national highways and expressways in Gansu Province were combined into a road layer, and the buffer analysis tool was used to produce the buffer zone of transportation elements and the buffer zone of administrative centers with 5km and 10km intervals as the center of the administrative areas of counties (districts), which were then superimposed on the analysis of the tourist attractions in Gansu Province in 2021, to derive the number of scenic spots in the buffer zones, respectively (Fig 8).

Fig 8 shows that for the transportation element, the number of tourist attractions gradually decreases as the buffer zone increases. There are 2991 tourist attractions in the buffer zone 5km from the transportation element, which is 73.1% of the total number of tourist attractions in Gansu Province and six times the number of tourist attractions in the 10km buffer zone. In addition, the tourist attractions show obvious distribution characteristics along the line, and are more densely distributed in the areas where the transportation routes intersect. The above analysis shows that transportation has a significant influence on the distribution of tourist attractions. Taking the county (district) level administrative center as the center of the circle and 30km as the radius, the number of tourist attractions in the buffer zone is 3,542, accounting for 86.6% of the number of tourist attractions in Gansu Province, among which there are 1,990 tourist attractions within the 10km buffer zone, accounting for 48.6% of the number of tourist attractions, 875 tourist attractions within the buffer zone of 10km to 20km, accounting for 21.4% of the number of tourist attractions, and 677 tourist attractions within the buffer zone of 20km to 30km, accounting for 16.5% of the number of tourist attractions. The number of tourist attractions within the buffer zone from 10km to 20km is 875, accounting for 21.4% of the total number of tourist attractions, and the number of tourist attractions within the buffer zone from 20km to 30km is 677, accounting for 16.5% of the total number of tourist attractions. The above results can show that the distribution of tourist attractions in Gansu Province takes the administrative center of each region as the core, and presents a scattered distribution pattern from dense to sparse within the circular buffer zone, which further supports that the spatial distribution of tourist attractions in Gansu Province has a close correlation with the regional location.

## 5.4 Tourist attractions and national policies and socio-economics

National policy is an important driving force for the healthy and sustainable development of the tourism industry, escorting the high-quality development of the tourism industry. The social economy creates a material foundation for tourism development, and the rapid development of tourism can also feed the regional economic growth, and there is a mutually beneficial relationship between the two. An OLS econometric model was used to measure the correlation between national policies and socio-economic factors and the number of tourist attractions in Gansu Province, and to explore the influence between the two. In this paper, the frequency of the words "tourism" and "attractions" appearing on the websites of municipal governments in Gansu Province is used to characterize the national policy factors, and the total income from tourism and the disposable income of residents are used to characterize the socio-economic factors. As shown in Table 4, the selected variables have a positive impact on the number of

**Table 4. Estimated results of the econometric model of the drivers of the spatial distribution pattern of tourist attractions in Gansu Province.**

| Variant | VIF [c] | Robust_Pr [b] | Standard deviation | probabilistic [b] |
|---|---|---|---|---|
| State policy | 1.083167 | 0.0548014 | 1.008878 | 0.685781 |
| Tourism revenue | 1.090744 | 0.0071836* | 354.967407 | 0.839893 |
| Disposable income | 1.072607 | 0.0632079 | 1.007294 | 0.644926 |

tourist attractions, with the most significant impact of total tourism income, and the impact of national policies and residents' disposable income on the number of tourist attractions decreasing in the order of magnitude.

# 6 Conclusion and discussion

## 6.1 Conclusion

Based on the POI point element data of tourist attractions in Gansu Province, this paper classifies the tourist attractions in Gansu Province into the categories of architectural facilities, geographic landscapes, biological landscapes, water sceneries, ruins and relics, and humanistic activities according to the classification standards of the national tourism resources. Combined with the use of spatial autocorrelation analysis and kernel density analysis in GIS spatial analysis method to analyze the spatial distribution pattern of various types of tourist attractions in Gansu Province, the driving factors in terms of socio-economics, transportation facilities, tourism services, natural environment, national policies and cultural background were analyzed in terms of their role mechanisms, and the driving mechanism of forming the spatial pattern of the tourist attractions was studied using spatial autoregressive model. The main conclusions are as follows.

1. Gansu Province has a large number and rich types of tourist attractions, and the number of different types of tourist attractions in different cities and towns varies to some extent. A total of 4092 POI data of tourist attractions in Gansu Province were obtained after data cleaning, among which the number of attractions in the category of building facilities was the largest, accounting for 67.5% of the total. Mainly due to the long history of Gansu Province and the large number of ethnic minorities, with the construction of a larger number of cultural landscape buildings and places of religious activity. Humanities activities tourist attractions in the province's tourist attractions in the least number of types, only 70, accounting for only 1.71% of the number of tourist attractions in the province, and the colorful cultural resources in Gansu Province features a mismatch phenomenon.

2. Tourist attractions in various cities and towns in Gansu Province show obvious clustering attributes in the remaining five categories of tourist attractions, except for the insignificant clustering trend in the category of ruins and relics. Among them, the formation of tourist attractions in the categories of human activities, biological landscapes and architectural facilities is more influenced by human factors, and Moran's I index is relatively high among the six categories of tourist attractions. Ruins and monuments of tourist attractions show random spatial characteristics, which is due to the large regional differences in the distribution of early human sites and monuments of ancient civilizations. In addition, the central and northwestern parts of the province have a higher degree of agglomeration, while the southwestern and southeastern corners have a lower degree of agglomeration, and the overall distribution is characterized by both agglomeration and dispersion.

3. The overall spatial distribution pattern of tourist attractions presents a multi-core decentralized distribution characteristics, forming eight core aggregation areas in the southeast. Among them, the capital city of Lanzhou has a high number and density of tourist attractions, followed by Tianshui City and Linxia Prefecture. The northwestern region is characterized by fragmentation, which is evident in parts of Dunhuang and Jiuquan cities.

4. The article analyzes the mechanism of action of the drivers at the level of national policy, natural environment, socio-economy, location and transportation, tourism services, and cultural context. And using buffer zone analysis to visualize the impact of the natural environment and locational traffic on the distribution of tourist attractions and to analyze the results. Using the OLS model to measure the relevant characterization indicators of socio-economic and national policies, the results show that all three selected characterization factors have a positive effect on the influence of the spatial distribution pattern of tourist attractions.

## 6.2 Discussion

In this paper, the spatial pattern of various types of tourist attractions in 14 prefectural and municipal cities in Gansu Province was first quantitatively analyzed by categorizing the crawled POI data of tourist attractions in Gansu Province. Secondly, the drivers of the spatial pattern of tourist attractions were analyzed both quantitatively and qualitatively. The analysis results show that there are big differences in the distribution of tourist attractions between cities and towns in Gansu Province, and the development of tourist attractions between regions is not balanced. Some municipalities have a high level of tourism development and the construction of tourist attractions has matured, while others have not given full play to their own advantages, and the construction of tourist attractions is relatively scarce. The number and spatial pattern of the 6 types of tourist attractions also show a non-equilibrium situation, the overall failure to grasp the advantages of the province's cultural resources, the integration of culture and tourism is still in the initial stage of exploration. Based on the above analysis, Gansu Province should seize the opportunity to take the background of the Belt and Road as an important driving force for the high-quality development of the tourism industry, enhance the level of construction of the tourism transportation network, strengthen the linkage of tourist attractions between the regions, and form a regional tourism loop, so as to take the point to bring the whole area, and promote the high-quality development of the regional economy and the tourism industry. Improve the construction of tourism infrastructure, improve the quality of public services, further enhance the travel experience of tourists, and synchronize to improve the rate of re-visit to attract more potential groups of tourists; finally, it should actively explore the advantages of its own resources, and innovate the development of culture and tourism fusion industry, so as to better release the vitality of tourist attractions.

## Author Contributions

**Data curation:** Wanqianrong Gao.

**Formal analysis:** Wanqianrong Gao.

**Project administration:** Ruijuan Peng.

**Resources:** Ruijuan Peng.

**Software:** Wanqianrong Gao.

**Supervision:** Ruijuan Peng.

**Writing – original draft:** Wanqianrong Gao.

**Writing – review & editing:** Ruijuan Peng.

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
