## [Decision Letter · Decision Letter 0]

13 May 2023

PONE-D-23-10108Spatial distribution pattern and driving mechanism of tourist attractions in Gansu Province based on POI dataPLOS ONE

Dear Gao,

Thank you for submitting your manuscript to PLOS ONE. After careful consideration, we feel that it has merit but does not fully meet PLOS ONE’s publication criteria as it currently stands. Therefore, we invite you to submit a revised version of the manuscript that addresses the points raised during the review process.

After reviewing the comments and suggestions provided by the reviewers, I recommend a Major Revision for your submission.

To ensure that your manuscript meets the necessary standards, it is crucial that you respond to each comment and suggestion from the editor and all reviewers in a comprehensive manner.

I would also like to provide some additional comments. The research question of your study is unclear. I suggest that you clarify these aspects of your research to improve the manuscript's overall quality.

Furthermore, several key pieces of literature on spatial autocorrelation analysis are missing. It is essential to include relevant and influential literature to establish the context of your research and demonstrate a comprehensive understanding of the subject matter.

Please ensure that you include the key literature, such as Anselin (1988) on Spatial Econometrics and Anselin (1995) on Local Indicators of Spatial Association (LISA).

Finally, I would like to mention that the language used in the paper is challenging to understand. There are several unnecessarily lengthy sentences that could be simplified for clarity.

We look forward to receiving your revised manuscript.

Kind regards,

Sutee Anantsuksomsri

Academic Editor

PLOS ONE

Journal Requirements:

2. Thank you for submitting the above manuscript to PLOS ONE. During our internal evaluation of the manuscript, we found significant text overlap between your submission and previous work in the [introduction, conclusion, etc.].

Please revise the manuscript to rephrase the duplicated text, cite your sources, and provide details as to how the current manuscript advances on previous work. Please note that further consideration is dependent on the submission of a manuscript that addresses these concerns about the overlap in text with published work.

[If the overlap is with the authors’ own works: Moreover, upon submission, authors must confirm that the manuscript, or any related manuscript, is not currently under consideration or accepted elsewhere. If related work has been submitted to PLOS ONE or elsewhere, authors must include a copy with the submitted article. Reviewers will be asked to comment on the overlap between related submissions (http://journals.plos.org/plosone/s/submission-guidelines#loc-related-manuscripts).]

We will carefully review your manuscript upon resubmission and further consideration of the manuscript is dependent on the text overlap being addressed in full. Please ensure that your revision is thorough as failure to address the concerns to our satisfaction may result in your submission not being considered further

3. We note that Figure 1, 2, 4, 5 in your submission contain [map/satellite] images which may be copyrighted. All PLOS content is published under the Creative Commons Attribution License (CC BY 4.0), which means that the manuscript, images, and Supporting Information files will be freely available online, and any third party is permitted to access, download, copy, distribute, and use these materials in any way, even commercially, with proper attribution. For these reasons, we cannot publish previously copyrighted maps or satellite images created using proprietary data, such as Google software (Google Maps, Street View, and Earth). For more information, see our copyright guidelines: http://journals.plos.org/plosone/s/licenses-and-copyright.

1. You may seek permission from the original copyright holder of Figure 1, 2, 4, 5  to publish the content specifically under the CC BY 4.0 license.  

4. Your abstract cannot contain citations. Please only include citations in the body text of the manuscript, and ensure that they remain in ascending numerical order on first mention.

Additional Editor Comments (if provided):

Thank you for submitting your manuscript, "Spatial distribution pattern and driving mechanism of tourist attractions in Gansu Province based on POI data," to PLOS ONE.

After reviewing the comments and suggestions provided by the reviewers, I recommend a Major Revision for your submission.

To ensure that your manuscript meets the necessary standards, it is crucial that you respond to each comment and suggestion from the editor and all reviewers in a comprehensive manner.

To make this process easier, please prepare a table that lists the points raised by each reviewer along with your corresponding responses. Additionally, highlight any changes made to the manuscript and include them in the rebuttal document.

I would also like to provide some additional comments. The research question of your study is unclear. I suggest that you clarify these aspects of your research to improve the manuscript's overall quality.

Furthermore, several key pieces of literature on spatial autocorrelation analysis are missing. It is essential to include relevant and influential literature to establish the context of your research and demonstrate a comprehensive understanding of the subject matter.

Please ensure that you include the key literature, such as Anselin (1988) on Spatial Econometrics and Anselin (1995) on Local Indicators of Spatial Association (LISA).

Finally, I would like to mention that the language used in the paper is challenging to understand. There are several unnecessarily lengthy sentences that could be simplified for clarity.

Reviewers' comments:

Reviewer's Responses to Questions

**Comments to the Author**

1. Is the manuscript technically sound, and do the data support the conclusions?

Reviewer #1: Yes

Reviewer #2: Yes

Reviewer #3: No

2. Has the statistical analysis been performed appropriately and rigorously? 

Reviewer #1: Yes

Reviewer #2: Yes

Reviewer #3: No

3. Have the authors made all data underlying the findings in their manuscript fully available?

Reviewer #1: Yes

Reviewer #2: Yes

Reviewer #3: No

4. Is the manuscript presented in an intelligible fashion and written in standard English?

Reviewer #1: Yes

Reviewer #2: Yes

Reviewer #3: No

5. Review Comments to the Author

Reviewer #1: 1. It is recommended to use Chinese standard maps with review numbers for maps involving national boundaries; 2. Try to use data from 2022 as much as possible; 3. For literature citations, please use English citation standards such as APA format according to the journal's requirements.

Reviewer #2: Based on the POI data, this paper analyzes the spatial distribution pattern of various types of tourist attractions with the spatial autocorrelation analysis and kernel density analysis methods in Gansu province, and probes into the driving mechanism. This study provides a scientific basis for the future development and layout of tourist developments and helps the tourism industry to develop with high quality in Gansu province. However, there are a number of major issues that the authors have to carefully consider in this manuscript:

1.The research significance or academic contribution of this article in the Introduction is insufficient. The authors only introduces the role and value of this study for Gansu province.

2.In the Part 3.1, the analysis focuses on the results of different methods rather than the spatial distribution of various types of tourist attractions. Moreover, this part only classifies the distribution of tourist attractions, without specifying the size and grade of the attractions, or explaining the causes of spatial distribution pattern.

3.In the Part 3.2.1, the analysis of these driving factors is only objective and general facts, which are weakly related to the spatial distribution of tourist attractions in Gansu, especially the natural environment level and level of transportation facilities. Some analyses and explanations are irrelevant or even wrong. For example, Gannan Prefecure and Jinchang City and other areas have not formed adequate road transportation system, and the tourist attractions are scattered. Does it imply that the scattered tourist attractions are due to poor road transportation system? In addition, it does not make sense that tourism service facilities and service personnel affect the formation of spatial distribution of tourist attractions.

4.In the Part 3.2.2, it is not logical to analyze the driving factors of the spatial distribution pattern of tourist attractions from the perspectives of social economy, transportation facilities and tourism services. The spatial distribution of tourist attractions, especially natural attractions, is the result of natural and geological action, and will not be more or less distributed due to social economy and tourism services. The star hotels, the number of tourists, and the number of accommodation and catering workers are not the driving factors of spatial distribution pattern of tourist attractions.

5.In the Part 3.2.3, GDP per capita, resident population, total road mileage, private car ownership and bus ownership are not the driving force of spatial distribution of tourist attractions.

In short, there are some major problems in the current paper, such as some deviation in its intention, insufficient logical argument, analysis of influencing factors instead of driving mechanism, etc., and the results and conclusions are not credible enough.

Reviewer #3: In this study, the analysis investigating the spatial influences applied inappropriate methods. Thus, the main findings of this study are overstated. This issue is critically considered as the serious flaw of this paper.

It is noted that conventionally, the spatial analysis employs spatial regression techniques (e.g., spatial lag model, spatial error model, spatial Durbin model) for quantifying the spatial spillovers. By integrating the GIS framework and the statistical inference, the spatial regression can systematically identify the relationship among variables on geographical space.

Some parts of this manuscript do not use English alphabets. It is inappropriate to mistakenly include non-English texts in the international publication.

6. PLOS authors have the option to publish the peer review history of their article (what does this mean?). If published, this will include your full peer review and any attached files.

Reviewer #1: No

Reviewer #2: No

Reviewer #3: No

---

## [Author Response · Author response to Decision Letter 0]

13 Aug 2023

Dear Editor,

We sincerely thank the editor and all reviewers for their valuable feedback that we have used to improve the quality of our manuscript. There reviewer comments are laid out below in italicized font and specific concerns have been numbered. Our response is given in normal font.

Sincerely,

Wanqianrong Gao, Master's degree

Tourism College of Northwest Normal University

730070, 13993813451, gaowanqianrong@163.com

Editor comments:

1.The research question of your study is unclear. I suggest that you clarify these aspects of your research to improve the manuscript's overall quality.

We sincerely thank the reviewer for careful reading. As suggested by the reviewer. We have sorted out the research questions and defined them more clearly. This is reflected in the blue font of the abstract and introduction.

2.Furthermore, several key pieces of literature on spatial autocorrelation analysis are missing. It is essential to include relevant and influential literature to establish the context of your research and demonstrate a comprehensive understanding of the subject matter. Please ensure that you include the key literature, such as Anselin (1988) on Spatial Econometrics and Anselin (1995) on Local Indicators of Spatial Association (LISA).

We sincerely appreciate the valuable comments. We have checked the literature carefully and added key literature on spatial econometrics and on local indicators of spatial associations to the methods section of the manuscript. (Reference 29, 30, 31, 32, 33, 34, 35)

3.Finally, I would like to mention that the language used in the paper is challenging to understand. There are several unnecessarily lengthy sentences that could be simplified for clarity.

Thanks for your suggestion. We have tried our best to polish the lan guage in the revised manuscript. Long sentences in the introduction and conclusion have been simplified.

4.During our internal evaluation of the manuscript, we found significant text overlap between your submission and previous work in the [introduction, conclusion, etc.].

We sincerely thank the reviewer for careful reading. We have revised the introduction and conclusion sections of the manuscript and introduced sources.

Review 1 Comments:

1.It is recommended to use Chinese standard maps with review numbers for maps involving national boundaries.

Thanks for your suggestion. We have carefully read the journal's graphic requirements and have completed the changes as required.

2.Try to use data from 2022 as much as possible.

Thanks for your suggestion. Our data was collected at the end of December 2021, and the paper was completed in 2022. As the data is the basis of the article, it is too much work to change the data, so we ask the reviewers to understand and sympathize.

3.For literature citations, please use English citation standards such as APA format according to the journal's requirements.

We were really sorry for our careless mistakes. Thank you for your reminder. We have changed the citation to the English literature citation standard.

Review 2 Comments:

1.The research significance or academic contribution of this article in the Introduction is insufficient. The authors only introduces the role and value of this study for Gansu province.

We sincerely thank the reviewer for careful reading. We have deepened the relevant elements, as reflected in the concluding part of the introduction.

2.In the Part 3.1, the analysis focuses on the results of different methods rather than the spatial distribution of various types of tourist attractions. Moreover, this part only classifies the distribution of tourist attractions, without specifying the size and grade of the attractions, or explaining the causes of spatial distribution pattern.

Thanks for your suggestion. We have revised the section to sort out the tourist attraction classes and explain the reasons for the spatial distribution pattern.

3.In the Part 3.2.1, the analysis of these driving factors is only objective and general facts, which are weakly related to the spatial distribution of tourist attractions in Gansu, especially the natural environment level and level of transportation facilities. Some analyses and explanations are irrelevant or even wrong. For example, Gannan Prefecure and Jinchang City and other areas have not formed adequate road transportation system, and the tourist attractions are scattered. Does it imply that the scattered tourist attractions are due to poor road transportation system? In addition, it does not make sense that tourism service facilities and service personnel affect the formation of spatial distribution of tourist attractions.

We sincerely appreciate the valuable comments. We have revised the section and re-selected the drivers that shape the spatial pattern of tourist attractions, which are now analyzed in terms of the natural environment, location and transportation, and socio-economics.

4.In the Part 3.2.2, it is not logical to analyze the driving factors of the spatial distribution pattern of tourist attractions from the perspectives of social economy, transportation facilities and tourism services. The spatial distribution of tourist attractions, especially natural attractions, is the result of natural and geological action, and will not be more or less distributed due to social economy and tourism services. The star hotels, the number of tourists, and the number of accommodation and catering workers are not the driving factors of spatial distribution pattern of tourist attractions.

We sincerely appreciate the valuable comments. We have made changes to the selection of drivers.

5.In the Part 3.2.3, GDP per capita, resident population, total road mileage, private car ownership and bus ownership are not the driving force of spatial distribution of tourist attractions.

We sincerely appreciate the valuable comments. We have removed the drivers you mentioned above from the manuscript.

Review 3 Comments:

In this study, the analysis investigating the spatial influences applied inappropriate methods. Thus, the main findings of this study are overstated. This issue is critically considered as the serious flaw of this paper. It is noted that conventionally, the spatial analysis employs spatial regression techniques (e.g., spatial lag model, spatial error model, spatial Durbin model) for quantifying the spatial spillovers. By integrating the GIS framework and the statistical inference, the spatial regression can systematically identify the relationship among variables on geographical space.

We sincerely appreciate the valuable comments. We changed the geodetector approach to quantitative analysis of drivers to OLS modeling in buffer analysis and spatial autoregressive analysis.

We tried our best to improve the manuscript and made some changes marked in blue in revised paper which will not influence the content and framework of the paper. We appreciate for Editors and Reviewers’warm work earnestly, and hope the correction will meetwith approval.Once again, thank you very much for your comments and suggestions.

---

## [Decision Letter · Decision Letter 1]

15 Sep 2023

Spatial distribution pattern and driving mechanism of tourist attractions in Gansu Province based on POI data

PONE-D-23-10108R1

Dear Dr. Gao,

We’re pleased to inform you that your manuscript has been judged scientifically suitable for publication and will be formally accepted for publication once it meets all outstanding technical requirements.

Kind regards,

Sutee Anantsuksomsri

Academic Editor

PLOS ONE

Additional Editor Comments (optional):

I have carefully reviewed the manuscript along with the comments of both reviewers.

The manuscript has been thoroughly revised and is now in good shape for the next publication process.

Reviewers' comments:

Reviewer's Responses to Questions

**Comments to the Author**

1. If the authors have adequately addressed your comments raised in a previous round of review and you feel that this manuscript is now acceptable for publication, you may indicate that here to bypass the “Comments to the Author” section, enter your conflict of interest statement in the “Confidential to Editor” section, and submit your "Accept" recommendation.

Reviewer #2: All comments have been addressed

2. Is the manuscript technically sound, and do the data support the conclusions?

Reviewer #2: Yes

3. Has the statistical analysis been performed appropriately and rigorously? 

Reviewer #2: Yes

4. Have the authors made all data underlying the findings in their manuscript fully available?

Reviewer #2: Yes

5. Is the manuscript presented in an intelligible fashion and written in standard English?

Reviewer #2: No

6. Review Comments to the Author

Reviewer #2: The authors have addressed the reviewer’s concerns and the quality of the manuscript has improved to some extent. But there are still some problems that the authors need to think about and deal with.

1.The research significance or academic contribution of this article is still insufficient. The value of this study is limited according to the current full text.

2.The logic of the paragraph needs to be carefully considered. For example, in the Section 3.1, the relationship between the first and second paragraph seems to be unclear. Those should be the data description. What is the characterization of the level and type structure?

3.The sentences and their explanation are perplexing. For example, from Lines 292 to 307, sentence repetition, sentence contradictions, and confusion in the presentation of county-level and city-level study units.

4.It seems to be unclear that the authors consider the mechanism of spatial distribution of tourist attractions. Part 3.3 is more likely to be the analysis of influencing factors than the analysis of mechanisms. The mechanism analysis in Part 4 is not closely related to the previous section.

5.It is suggested to adjust the structure of the paper by referring to English literature writing habits. For example, move the discussion section to the front of the conclusion section, and the mechanism analysis is including in the discussion section.

In short, this paper still needs to be well polished and improved before it can be considered for publication.

7. PLOS authors have the option to publish the peer review history of their article (what does this mean?). If published, this will include your full peer review and any attached files.

Reviewer #2: No

---

## [Editor Report · Acceptance letter]

25 Sep 2023

PONE-D-23-10108R1 

Spatial distribution pattern and driving mechanism of tourist attractions in Gansu Province based on POI data 

Dear Dr. Gao:

I'm pleased to inform you that your manuscript has been deemed suitable for publication in PLOS ONE. Congratulations! Your manuscript is now with our production department. 

Kind regards, 

on behalf of

Dr. Sutee Anantsuksomsri 

Academic Editor

PLOS ONE